# Carbohydrate Maldigestion and Intolerance

**DOI:** 10.3390/nu14091923

**Published:** 2022-05-04

**Authors:** Fernando Fernández-Bañares

**Affiliations:** 1Department of Gastroenterology, Hospital Universitary MútuaTerrassa, 08221 Terrassa, Spain; ffbanares@mutuaterrassa.es; 2Centro de Investigación Biomédica en Red de Enfermedades Hepáticas y Digestivas, Instituto de Salud Carlos III, 28029 Madrid, Spain

**Keywords:** sugar malabsorption, lactose, fructose, sorbitol, sucrose, FODMAP, irritable bowel syndrome, hydrogen breath test

## Abstract

This review summarizes dietary carbohydrate intolerance conditions and recent advances on the possible role of carbohydrate maldigestion and dietary outcomes in patients with functional bowel disease. When malabsorbed carbohydrates reach the colon, they are fermented by colonic bacteria, with the production of short-chain fatty acids and gas lowering colonic pH. The appearance of diarrhoea or symptoms of flatulence depends in part on the balance between the production and elimination of these fermentation products. Different studies have shown that there are no differences in the frequency of sugar malabsorption between patients with irritable bowel disease (IBS) and healthy controls; however, the severity of symptoms after a sugar challenge is higher in patients than in controls. A diet low in ‘Fermentable, Oligo-Di- and Monosaccharides and Polyols’ (FODMAPs) is an effective treatment for global symptoms and abdominal pain in IBS, but its implementation should be supervised by a trained dietitian. A ‘bottom-up’ approach to the low-FODMAP diet has been suggested to avoid an alteration of gut microbiota and nutritional status. Two approaches have been suggested in this regard: starting with only certain subgroups of the low-FODMAP diet based on dietary history or with a gluten-free diet.

## 1. Introduction

In the average Western diet, dietary starch provides around 60% of ingested absorbable carbohydrates; the disaccharides sucrose and lactose and the monosaccharide fructose are consumed to a lesser extent. Absorbable dietary carbohydrates must be broken down to their monosaccharide components before being transported across the surface membrane of the absorptive enterocytes. The process of digestion is completed by the action of disaccharide-specific hydrolases bound to the brush-border membrane of enterocytes lining the villi of the small intestine. The monosaccharide products of saccharidase hydrolysis (glucose and galactose) and the monosaccharides in the human diet (glucose and fructose) are ultimately transported into the enterocyte via specific membrane transport systems [1].

While, under normal conditions, most of an ingested carbohydrate load is completely absorbed before reaching the colon, several conditions can result in the impairment of absorption in the small intestine. Carbohydrate malabsorption can provoke an osmotically driven influx of fluid into the small bowel, leading to intestinal distension and rapid propulsion into the colon [2]. In addition, unabsorbed carbohydrates are rapidly fermented by colonic microbiota, generating gas, lactate, and short-chain fatty acids that have effects on gastrointestinal function (Figure 1) [3]. All of these factors can produce diarrhoea, gas, bloating, flatulence, and abdominal pain, which are symptoms that patients with carbohydrate intolerance usually report. Other factors involving the development of symptoms are: 1. the quantity and quality of the ingested carbohydrate load; 2. the rate of gastric emptying; 3. the response of the small intestine to an osmotic load; 4. gastrointestinal motility; 5. the metabolic capacity of the colonic microbiota; and 6. the compensatory capacity of the colon to reabsorb water and short-chain fatty acids [4].

This review summarizes carbohydrate intolerance conditions and recent clinical and basic science reports on the possible role of carbohydrate malabsorption and dietary outcomes in patients with functional bowel disease. Table 1 describes the mechanism of absorption of dietary carbohydrates and the enzyme drugs available to facilitate digestion.

## 2. Lactose Malabsorption and Intolerance

Recently, the different aspects of lactose malabsorption (LM) and its relationship to the development of clinical intolerance have been excellently reviewed [5,6], and the main points are summarized below.

LM is typically caused by lactase downregulation after infancy due to lactase non-persistence, which, in Caucasians, is mediated by the LCT−13910:C/C genotype [7]. A recent meta-analysis estimated the global prevalence of LM at 68%, with higher rates reported for genotyping data than hydrogen breath test (HBT) data [8]. Thus, lactase non-persistence cannot be considered a disease. Both lactase persistence and non-persistence are common phenotypes in healthy humans.

LM refers to an inefficient digestion of lactose in the small intestine. There are primary and secondary causes (such as viral gastroenteritis, giardiasis, and celiac disease).

Lactose intolerance is the occurrence of gastrointestinal symptoms such as bloating, borborygmi, flatulence, abdominal pain, and diarrhoea in patients with LM after ingestion of lactose. The difference between LM and lactose intolerance is important since a lactose-restricted diet is only indicated in patients with both malabsorption and intolerance [9].

Symptoms of lactose intolerance characteristically do not arise until there is less than 50% of lactase activity. However, since adults with lactase deficiency often maintain between 10% and 30% of intestinal lactase activity, symptoms only develop when they eat enough lactose to overcome the compensatory mechanisms of the colon. There appears to be a dose-dependent effect. A study of a Chinese population with proven lactase deficiency showed few symptoms at lower doses (10 g of lactose) and more symptoms at higher doses (20 g and 40 g) with an increase in a linear fashion [10]. When malabsorbed lactose reaches the colon, it is fermented by colonic bacteria, with the production of short-chain fatty acids (mainly acetate, propionate, and butyrate) and gas (hydrogen, carbon dioxide, and in some subjects, methane). The appearance of diarrhoea or symptoms of flatulence depends in part on the balance between the production and elimination of these fermentation products [2,11,12]. The short-chain fatty acids are rapidly absorbed by the colonic mucosa, favouring the concomitant absorption of water and electrolytes, and are an important energy fuel for the colonocyte and the organism. Diarrhoea occurs only under certain circumstances: (a) if the arrival speed of lactose to the colon exceeds the rate of bacterial fermentation of this sugar, which causes an osmotic overload in the colon and the appearance of diarrhoea; (b) if the capacity of bacterial fermentation in the colon is decreased (for instance, with the bacterial shift after the use of broad-spectrum antibiotics), which results in decreased production of short-chain fatty acids and, therefore, a lower capacity to absorb water and electrolytes; and (c) if there is a decrease in the absorption of short-chain fatty acids, as occurs in inflammatory diseases of the colon. Furthermore, gases produced by fermentation are consumed by the same bacteria or are absorbed and pass into the circulatory bloodstream [13]. For all these reasons, small amounts of lactose can be malabsorbed without inducing symptoms of intolerance.

HBT is the test of choice to assess LM and symptoms of intolerance [14,15,16]. Lactose HBT measures hydrogen produced by intestinal bacteria in the end-expiratory air after an oral challenge with a standard dose of lactose. In clinical practice, an intermediate lactose dose of 20–25 *g* in a 10% water solution is recommended [10,14,15,16].

As lactose intolerance is of greater clinical interest than LM, it has been debated whether it would be more relevant to use either ‘report of symptoms during a positive HBT’ or ‘disappearing of symptoms by a long-term lactose restricted diet after a positive HBT’ as a reference standard in diagnosis [9,17]. It is important to document that there is a relationship between carbohydrate intake and the occurrence of symptoms, and to consider that the symptoms are caused by the test carbohydrate [15]. Often, lactose malabsorbers do not develop symptoms after a challenge, and malabsorption without symptoms is not a major determinant for the outcome of the diet. In this sense, it has been suggested that the occurrence of symptoms during a lactose HBT strongly suggests a favourable response but does not help in predicting whether symptoms would subside or be reduced [18]. Conversely, their absence during the test was not associated with an acceptable negative predictive value. In another study, there were no significant differences in clinical improvement after a sugar-restricted diet between patients with either the presence or absence of clinical symptoms during HBT [19]. Thus, it is not as yet sufficiently clear whether the absence of symptoms after a positive HBT always gives an indication as to the role of sugar in the genesis of the patient’s symptoms. At present, therefore, lactose intolerance should be confirmed by a sustained, significant symptomatic relief of intestinal complaints after a lactose-free diet [9,15].

Reduction of lactose intake rather than exclusion is the key to the treatment of lactose intolerance [20]. In fact, while the intake of a large dose of lactose, of the order of 50 g of lactose (corresponding to one litre of milk), causes symptoms in most of the population with LM, most patients with lactose intolerance can ingest 12 g of lactose (equivalent to 200–250 mL of milk) with no symptoms, mainly when taken with other foods [21]. That is, the consumption of lactose with other foods likely slows gastric emptying and small intestinal transit, allowing lactose more time to be hydrolysed and absorbed. The role of lactase supplementation or the intake of the probiotics that produce lactase in the gut has been reviewed, and a positive effect was confirmed, though the effect was modest, and the quality of these studies was poor [5].

## 3. Sucrase-Isomaltase Deficiency

Sucrose, or saccharose, is commercially known as cane sugar or regular table sugar and consists of one glucose and one fructose molecule. The bond between these two molecules is broken by the membrane-bound enzyme sucrase-isomaltase. The same enzyme also hydrolyses the glucose molecules in the short oligosaccharides and starch [22]. Congenital sucrase-isomaltase deficiency (CSID) is a rare autosomal recessive condition with mutations of the sucrase-isomaltase gene on chromosome 3q25-26 [23]. Sucrase activity in intestinal villi is practically non-existent. The prevalence of CSID varies but has been described as 5–10% in Greenland, 3–7% in Canada, and 3% in Alaska. The prevalence in North America and Europe varies between 1/500 and 1/2000 [23]. Acquired forms of sucrase-isomaltase deficiency may be secondary to other chronic gastrointestinal conditions associated with intestinal villous atrophy, such as enteric infection, celiac disease, Crohn’s disease, and other enteropathies affecting the small intestine.

The symptoms usually appear in childhood and do not manifest until sucrose is included in the diet, which usually results from the introduction of fruit in the diet. It can also manifest at birth if a child is fed with a milk formula containing sucrose. In some individuals, it appears in adulthood with symptoms suggestive of IBS. In these cases, a careful anamnesis of the patient and their parents usually reveals a lifelong history of intestinal symptoms. Symptoms in childhood and in adulthood are similar, but the consequences are more serious in children. Severe watery diarrhoea may occur after the ingestion of small amounts of sucrose, which can be accompanied by abdominal pain, bloating, growth retardation, and the rejection of sugary foods. Isomaltose as such is not consumed in the diet. However, this oligosaccharide is released in the hydrolysis of starch. Although some patients may have mild symptoms after the ingestion of starch, most tolerate it well, mainly due to the low osmotic power of the molecule of undigested isomaltose.

Recent studies on patients with clinical symptoms suggestive of IBS, abdominal pain, diarrhoea, or bloating have shown the presence of sucrase-isomaltase deficiency after assessing disaccharidase activity in small bowel samples. Functional sucrase-isomaltose genetic variants appear to be more common in patients with symptoms suggestive of IBS than in controls and can lead to similar symptoms of maldigestion as those described in classical CSID [24,25,26]. This has the potential to identify groups among patients with IBS for individualized management. It seems to be a heterogeneous disorder, the severity of which is likely related to the biochemical phenotypes of the sucrase-isomaltase mutants, as well as the environment and diet of patients [27]. These studies suggest that the screening for sucrase-isomaltase mutations in patients with IBS may prove helpful when considering either a restrictive diet or enzyme replacement therapy as an appropriate treatment. However, genetic testing for sucrase-isomaltase mutations in individuals presenting with IBS-like symptoms is not routinely done in practice since it can be extremely costly.

A confirmatory CSID diagnosis can be performed by a disaccharidase assay using a small bowel tissue biopsy, genetic testing, or sucrose breath testing. The disaccharidase assay shows the levels of various enzymes such as sucrase-isomaltase, lactase, and maltase. Results are consistent with CSID if the amount of sucrose broken down by sucrase-isomaltase is lower than expected [28]. There are two available sucrose breath tests. The carbon-13 (^13^C) sucrose breath test involves the challenge with ^13^C-sucrose, and it is regarded to be the most direct and definitive measure for detecting CSID, with a sensitivity and specificity of 100% [29]. Additionally, the sucrose HBT may also be useful in diagnosing CSID [30], though many paediatric patients experience severe symptoms, passage of watery stools, bloating abdomen, and cramps from the 2 g/kg sucrose load. In contrast, this symptomatic response is not observed with the ^13^C-sucrose test because the load of sucrose ingested is only 0.02 g [29]. Genetic testing allows an unequivocal diagnosis in children with CSID. At least 80% of patients with CSID have one of four common mutations [31]. There are no comparative studies of these tests in adults with IBS-like symptoms who are carriers of functional sucrase-isomaltose genetic variants. A British Society of Gastroenterology review on the management of IBS stated that, at present, there is not enough evidence to consider routine testing for sucrase-isomaltase deficiency [32].

Most patients require a dietetic manipulation that, in general, must be stricter in childhood than in adult life. The degree of sucrose restriction necessary is different for each patient, who, by a trial-and-error method, becomes an expert in manipulating the diet to be symptom-free. It has been suggested that the use of sacrosidase (Sucraid^®^), an enzyme produced by Saccharomyces cerevisiae that hydrolyses sucrose, is effective as a treatment for sucrase-isomaltase deficiency. Double-blind studies have revealed that this enzyme, administered along with food, significantly prevents symptoms of intolerance in patients on a sucrose-containing diet as compared with placebo [33].

## 4. Fructose Malabsorption

Fructose is commonly obtained from sugar beets, sugar cane, and maize and is the sweetest of all natural sugars [34]. Fructose can be present as a monosaccharide or as disaccharide sucrose in a one-to-one molecular ratio with glucose. In recent decades, the solubility and sweetness of fructose have been exploited by the food industry in the form of sweetener blends derived from the enzymatic conversion of starches, with the increasing popularity of the use of high-fructose corn syrup (HFCS), a mixture of glucose and fructose in monosaccharide form, which may be of concern if it contains more fructose than glucose. In addition, use of fructose enhances the flavour and physical appeal of many foods and beverages. Thus, fructose is used in place of sucrose and other carbohydrates to reduce the caloric content of dietetic products while conserving high-quality sweetening profiles [35]. Between 1970 and 2004, the share of HFCS as a percentage of total sweetener used in the United States increased from half a percentage point to 42% [36]. The product is found in many beverages, including nearly all non-diet soda brands, as well as breakfast cereals, salad dressings, cheese spreads, yogurts, jams, peanut butter, and other foods.

HFCS is an American definition; in Europe, the term isoglucose or glucose–fructose syrup refers to a liquid sweetener composed of mainly glucose and fructose in varying compositions, which has a 20–30% fructose content compared to 42% (HFCS 42) and 55% (HFCS 55) in the United States [37]. The use of glucose–fructose syrup in soft drinks is limited in the European Union because manufacturers do not have a sufficient supply containing at least 42% fructose. As a result, soft drinks are primarily sweetened with sucrose, and the use of isoglucose as a replacement for sucrose in foods and beverages is not as widespread in Europe as in the United States. Data from Europe showed that daily free-fructose intake in adults was a mean of 17 g/d in Finland in 2007 [38], and a median of 15 g/d in the Netherlands in 2015 [39], far less than the 55 g/d reported for US adults in 2008 [40].

Fructose is mainly absorbed in the proximal small intestine. The absorption of monosaccharides is mainly mediated by the Na+-glucose cotransporter SGLT1 and the facilitative transporters GLUT2 and GLUT5 (reviewed in 1) (Figure 2). In brief, SGLT1 and GLUT2 are important for the absorption of glucose and galactose, while the GLUT5 transporter is related to fructose absorption. SGLT1 and GLUT5 are continuously localized in the apical brush-border membrane of enterocytes, whereas GLUT2 is localized in the basolateral membrane at low luminal glucose concentrations or in both the apical brush-border membrane and the basolateral membrane at high luminal glucose concentrations. Fructose monosaccharide is transported across the apical intestinal brush-border membrane via a Na+-independent facilitated diffusion mechanism via the GLUT5 transporter. GLUT5 expression is induced by fructose, which exerts a fast and strong upregulation of GLUT5 mRNA expression, leading to an increase in GLUT5 protein and activity levels. Approximately 90% of the fructose entering the enterocyte is metabolized, increasing the intracellular pool of glucose. The remaining fructose subsequently exits the enterocytes to enter the blood via the GLUT2 and GLUT5 transporters present at the basolateral membrane. GLUT2 may be recruited transiently in the apical brush-border membrane in response to high luminal glucose concentrations to assist in the absorption of excess luminal fructose. GLUT2 is a high-capacity, low-affinity glucose/galactose transporter that can co-transport fructose in a one-to-one ratio. GLUT2 is unable to transport fructose without the presence of glucose, although the mechanism for this is currently unknown.

Fructose malabsorption may be due to the insufficient uptake of fructose into enterocytes relative to the amount of luminal fructose. In addition, it may be caused by insufficient intracellular digestion of fructose, resulting in high intracellular fructose concentrations, which may contribute to decreased fructose uptake. In humans, fructose absorption capacity in the small intestine is much lower than glucose absorption capacity; it is very small after birth and increases later in response to dietary fructose. However, in combination with glucose, the capacity for fructose absorption increases due to the additional fructose uptake associated with Na^+^-glucose cotransport. Thus, fructose is well-absorbed in the presence of equimolar amounts of glucose in the proximal small intestine, while free fructose is absorbed slowly along the small intestine. Therefore, glucose co-ingestion significantly increases fructose absorption: glucose stimulates fructose absorption in a dose-dependent manner, and malabsorption occurs when fructose is present in excess of glucose. Unabsorbed fructose then passes into the colon and is fermented in the same manner as lactose in patients who have lactase non-persistence.

HBTs have been used to assess fructose malabsorption. However, there is insufficient information regarding the frequency of incomplete absorption of fructose in the healthy population. There are large individual variations in the ability to absorb fructose in healthy adults. Various studies have shown that the absorption capacity of fructose as a monosaccharide in healthy subjects ranges from less than 5 g to more than 50 g [41]. Thirty-seven to ninety percent of healthy subjects present malabsorption of a 50 g fructose load [41,42,43]. In addition, not only the dose, but also the solution concentration, affects fructose absorption. Incomplete absorption after a 50-g fructose load was observed in 37.5% (10% solution) and 71% (20% solution) of healthy people [42]. In a double-blind study in healthy subjects, increasing doses of fructose, 15 g, 25 g, and 50 g (at 10% solutions) were malabsorbed by 0%, 10%, and 80% of subjects. After a 50 g fructose load, 55% of subjects experienced symptoms of intolerance, which did not occur at lower doses [43]. In light of these results, it was claimed that healthy subjects have the capacity to absorb up to 25 g fructose. However, in a review of six studies, it was shown that 10% to 55% of healthy subjects did not absorb a 25 g fructose load, with either no symptoms or mild symptoms [44]. In addition, the ingestion of fructose as sucrose (50 to 100 g) did not result in appreciable malabsorption in these cases [41,42,45].

Neither is it clear whether fructose administration alone in fructose HBT reasonably reflects the conditions under which most dietary fructose is ingested. When fructose is ingested as part of a solid food, which delays gastric emptying and small intestinal transit, it does not reach the colon for at least 180 min; but when it is administered as an aqueous solution, it undergoes a fast and exponential gastric emptying, passing rapidly through the small intestine to reach the colon within 30 min [46]. In addition, the concomitant intake of sorbitol interferes with fructose absorption, whereas the concomitant ingestion of glucose in food enhances fructose absorption (via GLUT2).

The European guidelines on the indications, performance, and clinical impact of hydrogen and methane breath tests recommend that the dose of fructose in adults for the diagnosis of fructose malabsorption and intolerance should be 20–25 g [15]. However, the clinical utility of fructose HBT is debated. In fact, the Rome Consensus Conference on ‘Methodology and indications of H2-breath testing in gastrointestinal diseases’ stated that a fructose breath test is not recommended in clinical practice [14]. Moreover, the British Society of Gastroenterology guidelines for the investigation of chronic diarrhoea stated that, at present, fructose breath testing cannot be said to inform the diagnosis and treatment of fructose intolerance [47]. In a randomized control trial, it was observed that, despite the challenge of a relatively high dose of fructose (50 g) administered to reduce the risk for false-negative breath tests, 64.5% of patients with a normal fructose breath test improved with a fructose-restricted diet [48]. Thus, it is not clear that fructose HBT is an informative test regarding the response to a fructose-restricted diet since patients may improve on the diet despite a normal test and vice versa. Testing for fructose intolerance may replace testing for fructose malabsorption in the future as a way to obtain relevant clinical information. In another prospective randomized trial, patients with IBS were randomized for 12 weeks with/without a fructose-reduced diet in addition to a standard IBS diet [49]. In this study, the criteria for the diagnosis of fructose intolerance were based on improvement of self-reported symptoms while on a fructose-restricted diet and exacerbation of symptoms following a fructose challenge test. Using valid outcome measures to assess the fulfilment of these criteria, 56% of patients with IBS were considered to have fructose intolerance. Thus, there is the need to use standardized, validated symptom scales to obtain a reliable assessment of intestinal symptoms. In this sense, the ‘Carbohydrate Perception Questionnaire’, in both the paediatric and adult versions, seems to be a valid instrument for the assessment of symptoms developed after carbohydrate ingestion, with excellent psychometric properties [50].

The objective of a fructose-restricted diet is to restrict the intake of foods rich in fructose to a level that does not trigger intestinal symptoms. As mentioned, the main determinants of fructose malabsorption are the amount of fructose in excess of glucose and the intake of foods containing both fructose and sorbitol, since sorbitol interferes with fructose absorption. Patients are recommended to eliminate all foods with either an excess of free fructose or sorbitol (and other polyols) from the diet [19].

Xylose isomerase has been proposed as a potential treatment for fructose intolerance. The ability of xylose isomerase to convert between glucose and fructose has led to the suggestion of its use as a treatment for fructose intolerance. A double-blind, placebo-controlled study showed that oral administration of xylose isomerase was associated with a significant reduction in breath hydrogen after fructose ingestion, as well as a significant improvement in nausea and abdominal pain [51]. Further research is needed to assess the long-term health effects of xylose isomerase and to determine which patients may be most suitable for treatment.

## 5. Sorbitol Intolerance

Sugar alcohols, a class of low-molecular weight polyols, can occur naturally or be obtained by the hydrogenation of sugars. The most common are sorbitol, mannitol, maltitol, isomalt, lactitol, and xylitol. Sorbitol (D-glucitol) is the most frequently consumed sugar alcohol. Small amounts of sorbitol are present in some fruits of the Rosaceae family (apples, pears, cherries, apricots, peaches, and prunes). Dietary intake in the UK National Diet and Nutrition Survey data showed that the average daily intake of polyols was 3.5 g, with the 95th percentile at 10.4 g [52]. Most sorbitol intake comes from added sources [53]. Sorbitol contents in certain sugar-free sweet foods (e.g., sugar-free chewing gum, candy, mints, jam, diet drinks, chocolate) may be considerable. Sorbitol is also used as an additive for purposes other than sweetening in foods because of its unique combination of functional properties, including its role as a humectant, thickener, stabilizer, plasticizer, and emulsifier.

Sorbitol is poorly absorbed by the small intestine. It is well-known that, at high doses and concentrations, sorbitol is a laxative. Test solutions containing 10 g and 20 g resulted in 90% and 100%, respectively, of healthy volunteers showing malabsorption [54]. After a 5 g dose administered at concentrations of 2%, 4%, 8%, and 16%, malabsorption was manifested in 10%, 12%, 22%, and 43% of healthy volunteers [54]. Simultaneous ingestion of sorbitol and fructose seems to increase the malabsorption of the latter [55,56]. Thus, polyol restriction has been incorporated as a part of a FODMAP (acronym for Fermentable, Oligo-Di- and Monosaccharides and Polyols) diet [57] (see below).

It should be emphasized that, as with fructose HBT, sorbitol HBT should not be recommended in clinical practice for either adults or children [14].

## 6. Sugar Malabsorption and Functional Bowel Disease

Different studies have shown that there are no differences in the frequency of sugar malabsorption between patients with IBS and healthy controls, although the severity of symptoms after a sugar challenge is greater in patients than in controls [10,45]. In a single-blind randomized controlled study in patients with diarrhoea-predominant IBS and healthy subjects [45], sugar malabsorption was assessed by HBT after an oral load of various solutions containing lactose (50 g), fructose (25 g), sorbitol (5 g), fructose plus sorbitol (25 + 5 g), and sucrose (50 g). The frequency of sugar malabsorption was high in both patients and healthy controls, with malabsorption of at least one sugar solution in more than 90% of the subjects, but all subjects absorbed the sucrose solution. However, the symptoms score after both lactose and fructose plus sorbitol malabsorption was significantly higher in patients than in control subjects. In addition, more severe symptoms were observed in the IBS-D group after both lactose and fructose–sorbitol malabsorption than after the sucrose load administered as a control solution. Significantly more symptoms, although of mild intensity, were also observed after the sucrose load in patients with IBS than in healthy subjects. Finally, the administration of 10 g lactulose, a nonabsorbable carbohydrate, induced more symptoms and H_2_ production in patients with IBS than in healthy controls. In this study, 40–50% of both patients with IBS and healthy controls malabsorbed the 25 g fructose load, and the severity of symptoms was no different between patients and controls.

In another double-blind randomized study [10], patients with IBS-D and controls were given HBT to detect malabsorption and intolerance following the administration of 10, 20, and 40 g lactose. All participants had the lactase genotype C/C-13910, which is associated with lactase non-persistence. Malabsorption of 40 g lactose was observed in 93% of controls and 92% of IBS-D patients. Fewer controls than patients with IBS-D were intolerant to 10 g lactose, 20 g lactose, and 40 g lactose. The frequency of positive tests for lactose malabsorption and intolerance in both controls and IBS-D patients increased with the lactose dose, and the breath hydrogen excretion (peak and AUC H_2_ excretion) was associated with the severity of abdominal symptoms.

Therefore, the frequency of sugar intolerance after malabsorption seems to be higher in patients with IBS-D than in controls. The presence of a functional bowel disease increased the likelihood that an individual would report abdominal pain, bloating, and diarrhoea. In fact, symptoms experienced during breath testing, but not malabsorption, seem to correlate with previous functional bowel disease symptoms [58]. In that study, adequate symptom relief with dietary adaptation was achieved in >80% of intolerant patients, irrespective of malabsorption, which is a response rate similar to that in comparable studies [19,59,60], leading the way to a more comprehensive diet that reduces all sources of poorly digested, rapidly fermentable carbohydrates.

## 7. Carbohydrate-Reduced Diets in Functional Bowel Disease

More recently, there has been renewed interest in evaluating the role of FODMAPs in patients with IBS and carbohydrate-related symptoms [57]. FODMAPs are fermentable short-chain carbohydrates found in a variety of fruits, vegetables, pulses, dairy products, artificial sweeteners, and wheat. Evidence of a relationship between dietary FODMAPs and intestinal symptoms comes from a double-blind, cross-over study challenging patients with IBS with increasing doses of either glucose, fructose, fructans, or a mixture of the latter two for up to two weeks [61]. Fructose or fructans were significantly more likely than glucose to induce symptom recurrence. A further cross-over study found that patients developed fewer symptoms after a low-FODMAP diet compared with a typical Australian diet [62]. Afterwards, a randomized controlled trial found no differences between a low-FODMAP diet and an empiric IBS diet (NICE guidelines) based on healthy eating patterns, low fat content, and the avoidance of high-fibre food and resistant starch [63]. Current British Society of Gastroenterology guidelines on the management of IBS recommend that a diet low in FODMAPs, which is considered to be a second-line dietary therapy, is an effective treatment for global symptoms and abdominal pain in IBS, although its implementation should be supervised by a trained dietitian, and fermentable oligosaccharides, disaccharides and monosaccharides, and polyols should be reintroduced according to tolerance (recommendation: weak, quality of evidence very low) [32].

The low-FODMAP diet is designed as a three-phase approach, whereby patients restrict all FODMAP subgroups for a 6–8 week period, followed by a reintroduction phase for the re-challenge of subgroups to test tolerance, followed by a personalised long-term maintenance phase with periodic re-challenge of poorly tolerated foods [64], in a ‘top-down’ approach. In addition, a ‘bottom-up’ approach has been suggested, starting with the restriction of a few specific foods or FODMAP subgroups based on baseline diet history and patient-reported triggers [46] (Figure 3). This ‘bottom-up’ approach has been promoted to avoid prolonged dietary restrictions on a low-FODMAP diet, potentially preventing the disruption of the gut microbiota and micronutrient status. In addition, it has been suggested that a gluten-free diet may be the easiest way of achieving fructan reduction [65], since fructans are a key component to be reduced in a long-term adapted low-FODMAP diet, as demonstrated in a prospective study [66]. In this sense, it has been proposed that a gluten-free diet may be administered as a ‘bottom-up’ approach in the FODMAP diet for patients with functional bowel disease [67,68,69]. In addition, patients have regarded a gluten-free diet as more acceptable than a low-FODMAP diet [70]. In fact, only 40% of patients show a good adherence to the low-FODMAP diet [71].

Twenty to fifty percent of IBS patients do not experience a reduction in gastrointestinal symptoms when following NICE guidelines and/or a low-FODMAP diet [64]. Other dietary approaches have been suggested. In a recent study that randomized patients with IBS to a 4-week starch- and sucrose-reduced diet or to a habitual diet (control group) [72], the intervention group presented a significantly lower total IBS-SSS, ‘abdominal pain’, ‘bloating/flatulence’, and ‘intestinal symptoms influence on daily life’ scores compared to controls, and a 74% response rate. Although promising, this dietary treatment needs to be further evaluated and compared to establish dietary treatments before it can be routinely used in a clinical setting.

## 8. Mechanisms of Carbohydrate-Reduced Diet Improvement in Functional Bowel Disease

The mechanism by which dietary changes may affect symptoms in IBS patients was explored in a cross-over trial in which patients with IBS and bloating were recruited alongside a parallel cohort of healthy volunteers without functional gastrointestinal symptoms who followed the same trial regimen [73]. Subjects were given 40 g of carbohydrate (glucose, fructose, and inulin in random order) in a 500-mL solution. Levels of breath hydrogen were measured, and intestinal content was assessed by MRI before and at various time points after the consumption of each drink. IBS patients and healthy subjects had similar physiological responses following fructose or inulin ingestion. These results indicate that colonic hypersensitivity to distension, rather than excessive gas production, produces the carbohydrate-related symptoms in IBS patients. Zhu et al. [74] reported on lactose responsiveness in a Chinese population with a high prevalence of lactose maldigestion. They included IBS patients and healthy controls who underwent a 20 g lactose HBT, with assessments of hydrogen gas production and lactose intolerance symptoms. Lactose intolerance was more frequent in IBS than in healthy controls, especially bloating and borborygmus. Rectal hypersensitivity assessed by barostat was associated with a higher odds ratio of bloating than hydrogen production, suggesting that visceral hypersensitivity plays an important role in carbohydrate intolerance in IBS.

Therefore, osmotically active unabsorbed monosaccharides and disaccharides distend the small bowel with fluid and, subsequently, the colon, where they produce a gas increase and, in those subjects with visceral hypersensitivity, induce more severe gastrointestinal symptoms. Furthermore, the rapid colonic fermentation of unabsorbed carbohydrates generates gas and produces short-chain fatty acids, which lower colonic pH and trigger bowel symptoms [75]. When delivered as liquid drinks, they speed gastric emptying, and the increase in small-bowel water content also accelerates intestinal transit, reducing small-bowel absorption, which may make symptoms more severe. Evidence that there are differences in visceral hypersensitivity in subsets of IBS patients suggests that the same magnitude of stimulus will produce different degrees of symptom response in patients depending on their sensory threshold [32,76]. In the case of carbohydrate malabsorbers without IBS, symptom generation may be mainly triggered by rapid colonic fermentation.

A role for barrier dysfunction and inflammation produced by a high-FODMAP diet has been proposed. In animal models, there is an association between fructose or high-FODMAP intake, increased intestinal permeability, barrier dysfunction, and inflammation [77,78]. Thus, changing carbohydrate intake may induce barrier dysfunction, which may, at least in part, be driven by the colonic microbiota. In addition, in humans, changes in histamine levels suggest that increased FODMAP intake may influence immune activation [79]. Moreover, reductions in pro-inflammatory cytokines following a low-FODMAP diet support the immune modulation hypothesis [80].

## 9. Final Outlook

The development of a well-accepted, practical, and cost-effective carbohydrate intolerance test capable of predicting the outcome of dietary management is one of the major clinical challenges in the field of functional bowel disease. In this regard, more studies are needed to explore the role of fructose intolerance testing, with or without concomitant measures of H_2_ and CH_4_ excretion, using standardized and validated symptom scales. The role of screening for sucrase-isomaltase mutations for guiding the development of a rational treatment with sucrose-reduced diets in a subgroup of IBS patients appears promising, but also requires further study. The ‘bottom-up’ approach of a low-FODMAP diet should also be further evaluated. It is likely that many patients may benefit from this approach, thereby avoiding a long-term, highly carbohydrate-restrictive diet and preventing an alteration of gut microbiota and nutritional status. Two approaches have been suggested in this regard, starting with only some subgroups of the low-FODMAP diet based on dietary history or with a gluten-free diet. Further studies should provide high-quality evidence to document the clinical response and the long-term effects of these strategies.

## Figures and Tables

**Figure 1 nutrients-14-01923-f001:**
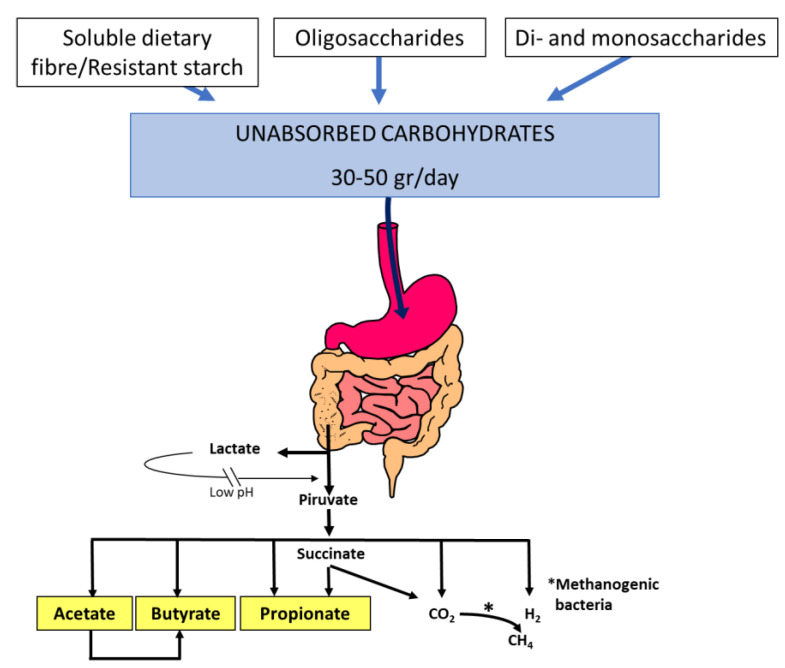
Short-chain fatty acid and gas (H_2_, CO_2_, CH_4_) production by the colonic microbiota fermentation of dietary unabsorbed carbohydrates. Thirty to fifty grams of soluble dietary fibre, resistant starch, oligosaccharides (fructo-oligosaccharides, galacto-oligosaccharides, raffinose, and stachyose), disaccharides (lactose, sucrose), and monosaccharides (fructose) enter the colon each day and become available for colonic fermentation by the microbiota. Acetate (C2), propionate (C3), and butyrate (C4) are the main short-chain fatty acids that play important roles in gastrointestinal function.

**Figure 2 nutrients-14-01923-f002:**
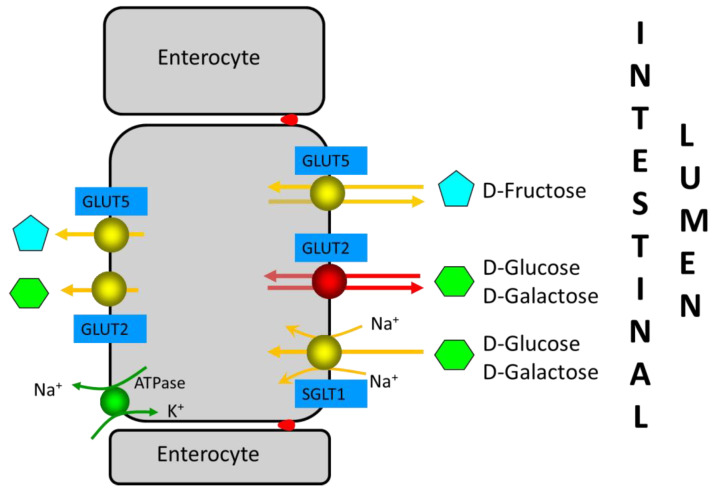
Enterocyte monosaccharide transporters involved in D-glucose, D-galactose, and D-fructose absorption in the small intestine. GLUT2, which is only observed in the apical brush-border membrane at high D-glucose concentrations in intestinal lumen, is indicated in red (see reference [1]). Red dots between enterocytes represent the tight junctions.

**Figure 3 nutrients-14-01923-f003:**
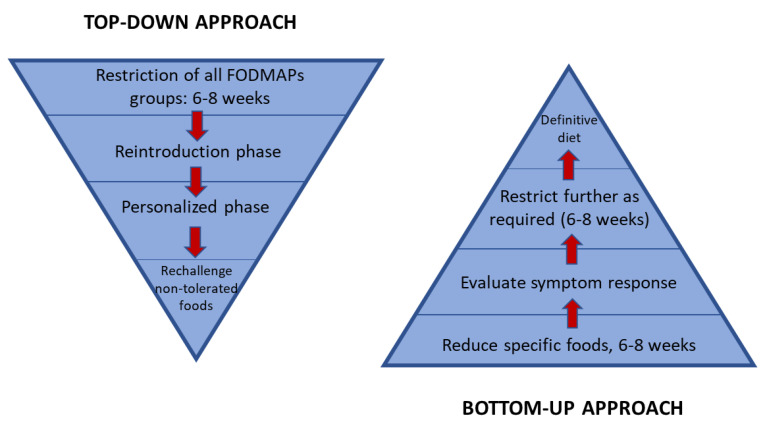
‘Top-down’ and ‘bottom-up’ approach to the low-FODMAP diet (adapted with permission from reference [46]; Copyright 2019, John Wiley & Sons, Inc). Reducing a few specific foods/subgroups in the ‘bottom-up’ approach implies an adequate diet history, for example, that in a patient who consumes chewing gum and/or candies, these are withdrawn; if he/she consumes large quantities of fruits and fruit juices, they may benefit from the restriction of only excess fructose and polyols to start with; or if he/she consumes large amounts of wheat, onion, artichokes, and pulses, they may be more likely to benefit from a restriction of fructans. Furthermore, in the ‘bottom-up’ approach, a gluten-free diet has been claimed to be the easiest way to reduce fructan intake (see text).

**Table 1 nutrients-14-01923-t001:** Absorption mechanisms of dietary carbohydrates and specific drugs available to facilitate digestion.

Carbohydrate	Type	Absorption Mechanisms	Available Specific Drugs
Monosaccharides	Fructose	Absorption of excess fructose occurs in the small intestine: rapidly via GLUT-2, the sodium-dependent active transport mechanism in conjunction with glucose; slowly via GLUT-5, a specific transporter for fructose using carrier-mediated facilitated diffusion. Thus, fructose is well-absorbed in the presence of equimolar glucose in the proximal small intestine, whereas free fructose is absorbed slowly along the length of the small intestine.	Xylose-isomerase
Disaccharides	Lactose Sucrose	Unabsorbed in small intestine if lactase is absent. Unabsorbed in small intestine in case of sucrase-isomaltase deficiency	Lactase Sacrosidase
Oligosaccharides	FOS * GOS	Humans do not possess small intestinal hydrolases to hydrolyse oligosaccharides, and they are unabsorbed.	None
Polyols	Sorbitol Mannitol Maltitol Isomalt Lactitol Xylitol	Sugar alcohols are poorly absorbed along the length of the small intestine by slow passive diffusion.	None

* FOS: Fructo-oligosaccharides; GOS: Galacto-oligosaccharides.

## Data Availability

Not applicable.

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
