# Peer review of "Carbohydrate Maldigestion and Intolerance"

_nutrients, 2022, doi:10.3390/nu14091923_

Round 1

Reviewer 1 Report

This is an interesting review, that summarizes carbohydrate intolerance conditions and recent clinical and basic science reports on the possible role of carbohydrate malabsorption and dietary outcomes in patients with functional bowel disease.

Sugar malabsorption can affect the severity of symptoms in patients with irritable bowel disease (IBS). A diet low in Fermentable, Oligo- Di- and Monosaccharides and Polyols (FODMAPs) could be a promising treatment for global symptoms and abdominal pain in IBS. Further studies in this field are needed.

The manuscript is well written and well structured. The topic is interesting and in line with the journal.

Author Response

Thanks for your comments

Reviewer 2 Report

The author reviews carbohydrate absorption-related issues with clinical relevance to bowel disease. The overall details are thorough about different carbohydrates and their absorption with clinical correlation with IBS.

The manuscripts will possibly improve with mechanistic details related to inflammation following carbohydrate malabsorption. The author can briefly discuss the current treatment modalities, mainly focusing on available medication or development. Is it possible to discuss the role of antibiotics related to malabsorption?

The author should include a table of the critical points such as carbohydrate name, absorption location in the digestive tract, significant transporters, clinical relevance, and drugs.

In figure 2: What are the red dots between enterocytes? Are the glucose transporters on the villi? It can be shown or mentioned.

Author Response

I thank the reviewer comments.

I have dealt with them as follows:

  1. The manuscript will possibly improve with mechanistic details related to inflammation following carbohydrate malabsorption. Answer: I have included a paragraph about this topic (page 12, lines 506 to 513).
  2. The author can briefly discuss the current treatment modalities, mainly focusing on available medication or development. Is it possible to discuss the role of antibiotics related to malabsorption? Answer: The main current treatment modality is dietary management and this has been extensively discussed. Available drugs to facilitate digestion of lactose, fructose and sucrose are mentioned in text (lactase, lines 143-146; sacrosidase, lines 209-213; and xylose isomerase, lines 342-349). Antibiotics are used to treat small-intestine bacterial overgrowth which is out of the focus of the present review. Broad-spectrum antibiotics could cause diarrhoea after carbohydrate malabsorption due to a decrease in the capacity of bacterial fermentation in the colon, as mentioned in text (lines 106 to 109).
  3. The author should include a table of the critical points such as carbohydrate name, absorption location in the digestive tract, significant transporters, clinical relevance, and drugs. Answer: I have included a Table as suggested by the referee.
  4. In figure 2: What are the red dots between enterocytes? Are the glucose transporters on the villi? It can be shown or mentioned. Answer: The red dots between enterocytes represent the tight junctions. This has been added.